# Antibacterial Effect of Lemongrass (*Cymbopogon*
*citratus*) against the Aetiological Agents of Pitted Keratolyis

**DOI:** 10.3390/molecules27041423

**Published:** 2022-02-19

**Authors:** Bettina Schweitzer, Viktória Lilla Balázs, Szilárd Molnár, Bernadett Szögi-Tatár, Andrea Böszörményi, Tamás Palkovics, Györgyi Horváth, György Schneider

**Affiliations:** 1Department of Medical Microbiology and Immunology, Medical School, University of Pécs, Szigeti út 12, H-7624 Pécs, Hungary; scbtabt.pte@pte.hu (B.S.); palkovics.tamas@pte.hu (T.P.); 2Department of Pharmacognosy, University of Pécs, Rókus u. 2, H-7624 Pécs, Hungary; balazsviktorialilla@gmail.com (V.L.B.); horvath.gyorgyi@gytk.pte.hu (G.H.); 3Research Institute for Viticulture and Oenology, University of Pécs, Pázmány Péter u. 4, H-7634 Pécs, Hungary; molnar.szilard@freemail.hu; 4Department of Pharmacognosy, Semmelweis University, Üllői u. 26., H-1085 Budapest, Hungary; szogi-tatar.bernadett@pharma.semmelweis-univ.hu (B.S.-T.); boszormenyi.andrea@pharma.semmelweis-univ.hu (A.B.)

**Keywords:** lemongrass, antibacterial, gas chromatography, bioautography, solid-phase microextraction, compound identification, pitted keratolysis

## Abstract

Pitted keratolysis (PK) is a bacterial skin infection mostly affecting the pressure-bearing areas of the soles, causing unpleasant symptoms. Antibiotics are used for therapy, but the emergence of antiobiotic resistance, makes the application of novel topical therapeutic agents necessary. The antibacterial effects of 12 EOs were compared in the first part of this study against the three known aetiological agents of PK (*Kytococcus sedentarius*, *Dermatophilus congolensis* and *Bacillus thuringiensis*). The results of the minimal inhibitory concentration, minimal bactericidal concentration and spore-formation inhibition tests revealed that lemongrass was the most effective EO against all three bacterium species and was therefore chosen for further analysis. Seventeen compounds were identified with solid-phase microextraction followed by gas chromatography–mass spectrometry (HS-SPME/GC-MS) analysis while thin-layer chromatography combined with direct bioautography (TLC-BD) was used to detect the presence of antibacterially active compounds. Citral showed a characteristic spot at the Rf value of 0.47, while the HS-SPME/GC-MS analysis of an unknown spot with strong antibacterial activity revealed the presence of α-terpineol, γ-cadinene and calamenene. Of these, α-terpineol was confirmed to possess an antimicrobial effect on all three bacterium species associated with PK. Our study supports the hypothesis that, based on their spectrum, EO-based formulations have potent antibacterial effects against PK and warrant further investigation as topical therapeutics.

## 1. Introduction

Pitted keratolysis (PK) is an acquired, chronic, non-inflammatory, superficial bacterial infection of the skin, confined to the stratum corneum of the soles. It is clinically characterized by multifocal, discrete, superficial crateriform pits and superficial erosions usually on the pressure-bearing areas on the soles of the feet, and rarely on the palms [1]. PK can affect patients of any age. Typical symptoms are described in healthy people wearing occlusive shoes, for example, athletes, sailors and soldiers [2]; aged, hospitalized patients [3]; and citizens of flooded areas, mostly from developing, under-resourced communities in humid, tropical countries [4]. Other predisposing factors include hyperhydrosis, elevated skin pH, poor foot hygiene, obesity, and immunodeficiency [5]. Malodour, the most characteristic symptom, is caused by sulfur compounds (thiols, sulfides, and thioesters) produced by the protease enzymes of the colonizing bacteria [5]. *Kytococcus* (formerly *Micrococcus*) *sedentarius* and *Dermatophilus congolensis* were documented as aetiological agents of PK [6,7], and in a very recent study the aetiological role of the Gram-positive, spore-forming *Bacillus thuringiensis* was demonstrated [8].

Today, the treatment of PK is primarly empiric, and is based on antibiotics, such as erythromycin, tetracycline and clindamycin. However, the spread of antibiotic resistance among the bacterial isolates limits therapeutic success [9].

Due to their antibacterial features, the use of essential oils (EOs) is a potential way to combat emerging antibiotic resistance [10]. EOs are secondary metabolites of aromatic plants consisting of ~20–60 components at different concentrations. There can be major compounds at relatively high concentrations (20–70%) compared to other components present in trace amounts [11].

*Cymbopogon citratus* (DC.) Stapf, commonly known as lemongrass, is a coarse grass with a strong lemon taste. It is a perennial herb widely cultivated in the tropics and sub-tropics [12]. Researchers have found that lemongrass holds antidepressant, antioxidant, antiseptic, astringent, nervine and sedative properties [13]. Furthermore, the antibacterial activity of lemongrass oil against a diverse range of species comprising Gram-positive and Gram-negative organisms, along with yeast and fungi, was formerly established [13,14,15]. The major component of lemongrass EO is citral, consisting of the two geometric isomers citral A (geranial) and citral B (neral) [16]. Other compounds identified in lemongrass EO include limonene, citronellal, β-myrcene and geraniol [16], along with trace compounds [17]. On the compound level, citral, the major component of lemongrass EO, is known to be antibacterial, but other compounds have not yet been studied in detail from this perspective. Furthermore, an earlier study has also reported about the capacity of thyme EO, cinnamon EO and lemongrass EO to reduce viable spores [18], which could be an important feature of these EOs, as *B. thuringiensis*, one aetiological agent of pitted keratolysis, is a typical spore-forming bacterium species, and spore fomation can influence survival and therefore therapeutic success [19].

The aim of this study was to screen the antibacterial potential of certain EOs against all the known aetiological agents of pitted keratolysis and try to reveal the most effective EO from an antibacterial point of view, including growth- and spore-formation inhibitions and bactericid effects. To best of our knowledge, this is the first study where these aspects were investigated in relation to the aetiological agents of PK, by combining antibacterial screenings, such as the drop plate method, minimal inhibitory and minimal bactericidal concentration determinations, spore-formation suppression test and analytic methods such as (TLC-DB) and headspace solid-phase microextraction followed by gas chromatography–mass spectrometry (HS-SPME/GC-MS) and thin-layer chromatography combined with direct bioautography (TLC-DB).

## 2. Results

### 2.1. Antibacterial Effects of EOs—Drop Plate Method

The antibacterial efficacy of 12 EOs showed marked differences using the drop plate method (Figure 1). Based on the diameter (20–36 mm) of the inhibition zones, thyme, cinnamon and lemongrass were the most effective antibacterial EOs against all three investigated bacterium species. Other EOs differed in their efficacies against the different species. Lemon was effective only against *D. congolensis* while salvia only against *K. sedentarius* and *D. congolensis*, and eucalyptus against *B. thuringiensis* and *D. congolensis.* From the bacterial point of view, the most sensitive bacterium was *D. congolensis*, as all EOs had some inhibitory effect on this species (Figure 1).

### 2.2. Minimal Inhibitory and Minimal Bactericidal Concentration Determinations (MIC and MBC)

The antibacterial effects of EOs were also tested in a liquid medium in order to determine their MICs and MBCs (Table 1). In the macrodilution tests, all EOs showed bactericidal effects at different concentrations against all three bacterium species.

### 2.3. Spore-Formation Inhibition Test

No bacterial growth was visually detected after 24 h incubation when the applied lemongrass EO concentrations in the case of the *B. thuringiensis* suspensions were above 0.2 mg/mL (MIC value). Transferring 1 mL from the optically clear tubes above the MIC values into a final volume of 25 mL of the medium, subsequent incubation (24 h at 37 °C) and confirmative outplating revealed that no living bacterium cell was detected in the presence of lemongrass EO at an original concentration of >12.8 mg/mL. In all other cases, no restrictive effect was revealed as the solutions became turbid after the 24 h incubation.

### 2.4. HS-SPME/GC-MS Analysis

Aromax Ltd.’s lemongrass EO contained 60.6% citral, 34.5% geranial and 26.1% neral. Its minor volatile compounds were caryophyllene-oxide (4.7%), geraniol (4.3%), linalool (2.6%), α-terpineol (1.7%), γ-cadinene (1.4%) and limonene (1.3%) (Figure 2, Table 2).

### 2.5. Analysis of Antibacterial Compounds of Lemongrass EO Using Thin-Layer Chromatography Combined with Direct Bioautography

The TLC-DB method was used to reveal the antibacterial components of lemongrass EO. According to the Rf value, the main component of lemongrass EO was citral. This was also confirmed by the commercial citral control. Based on its Rf value, we could not determine the identity of one characteristic spot. This spot was cut out (Figure 3) and analysed further with a preparative chromatographic method.

### 2.6. Component Identification with Headspace Solid-Phase Microextraction and Gas Cromatography–Mass Spectrometry (HS-SPME/GC-MS)

As a result of the HS-SPME/GC-MS analysis of the unidentified cut spot isolated from the TLC-bioautography plate, we were able to distinguish nine characteristic peaks and several minor ones (Figure 4). α-Terpineol, γ-cadinene and calamenene abundance were 13.2%, 7.0% and 3.7%, respectively.

### 2.7. Antibacterial Effects of Citral and α-Terpineol: Drop Plate Method

Characteristic differences could be observed between the antibacterial activities of citral and α-terpineol using the drop plate method. Citral produced larger clearing zones by diameter with each bacterium species compared to α-terpineol (Table 3). The differences were the most pronounced in *D. congolensis* and *B. thuringiensis*, and were minimal with *K. sedentarius*.

### 2.8. MIC and MBC of Citral and α-Terpineol

Macrodilution-based antibacterial testing of the citral and α-terpineol revealed that α-terpineol showed weaker antibacterial effects in the applied test range (0.1–12.8 mg/mL) (Table 4). In contrast, citral strongly inhibited the proliferation of all three Gram-positive bacterium species at low MIC and MBC values.

### 2.9. Confirmation the Antibacterial Features of Citral and α-Terpineol with Thin-Layer Chromatography Combined with Direct Bioautography (TLC-DB)

Qualities of the chromatographic separations on the thin-layer silica plates were confirmed before developments by illuminating the TLC plates under UV light (Figure 5A and Figure 6A). Here, some major components or mass of components became visible. Vanilin sulfuric acid developments of one plate from each parallel run revealed the presence of further compounds in the lemongrass EO sample (Figure 5B(1) and Figure 6B(1)). The TLC-based direct bioautography method showed another antibacterially active compound or compound group that was effective against all three tested bacteria (Figure 5C,D and Figure 6C), but was definitely separated in the case of *K. sedentarius* (Figure 5D(1)). In the cases of *B. thuringiensis* and *D. congolensis* (Figure 5C(1) and Figure 6C(1)), the presenting components of lemongrass EO showed a smeared outlook in contrast to *K. sedentarius*.

Lemongrass EO was diluted for application to *B. thuringiensis* and *K. sedentarius,* but not with *D. congolensis*. Citral (Rf = 0.47) and α-terpineol (Rf = 0.36) were active against both bacteria (Figure 5C,D and Figure 6C).

## 3. Discussion

In the present study the antibacterial effects of lemongrass EO against three bacterium species associated with PK were investigated. As a superficial skin infection, PK is an easily accessible target for topical medications, and EO-based gels and ointments are potential therapeutic candidates [20]. The 12 EOs for our antibacterial study were chosen based on earlier publications reporting their antibacterial features [21,22,23,24,25,26,27,28,29,30]. No data were available concerning the sensitivities of the known aetiological agents of PK (*B. thuringiensis*, *Kytococcus sedentarius* and *Dermatophilus congolensis*).

The results of the drop plate method performed with the three tested bacterium species indicated that although the majority of the 12 EOs were antibacterial, this effect was most pronounced in the cases of lemongrass, cinnamon and thyme (Figure 1). The antibacterial potential of the tested EOs was also supported by the MIC and MBC determinations performed in the macrodilution tests, although there were some discrepancies in their efficacies. This may be due to the differing characteristics of the two test methods. The large inhibition zones (>25 mm) and low MIC values (0.1 mg/mL) recorded here related to thyme EO confirm the findings of previous authors [31]. Its strong antimicrobial effect is attributed to its high phenolic compound content, including thymol and carvacrol, which constitute > 40% of this oil [32]. Kačániová et al. have previously shown that cinnamon EO was very effective against *B. subtilis*, where the MIC value was 0.10 mg/mL, similarl to our results in the cases of all the tested bacteria [33]. In this case, cinnamaldehyde and eugenol, the major components of cinnamon EO, proved to be responsible for the antibacterial activity [34]. Similar to this recent work, the antibacterial role of one major EO component was demonstrated by our results using lemongrass EO and citral. We have also attributed the bactericidal effects to at least one other lemongrass EO component (α-terpineol). Further studies are required to clarify the roles of cadinene and calamenene. Altogether, the low inhibitory concentration (0.13 mg/mL) of lemongrass EO against a *Bacillus* species, *B. cereus* [35], was also in harmony with our findings on *B. thuringiensis*.

From among the most effective bactericidal EOs (thyme, cinnamon and lemongrass), lemongrass was chosen for further studies because of the following practical considerations: *B. thuringiensis*—a recently identified causative agent of PK—is a spore-forming bacterium. Although a recent study has shown that cinnamon, thyme, and lemongrass EOs were all able to reduce the number of viable spores, this effect was the most pronounced in the case of lemongrass EO [18,36]. Furthermore, cinnamon could cause contact dermatitis if exposed to the skin surface [37].

Percentage-based compound composition results, obtained by HS-SPME/GC-MS analysis, were in agreement with previous studies [38,39,40]. A former study has reported that West Indian lemongrass EO (Cymbopogon citratus (DC.) Stapf.) contains about 77% citral. Natural citral is a mixture of two geometric isomers: geranial (citral a, €- or α-citral) and neral (citral b, (Z)- or β-citral). In the case of West Indian lemongrass EO, their percentage distribution is usually as follows: geranial 36.7–55.9% and neral 25.0–35.2% [41]. Similarly, our results showed that neral comprised 26.1% and geranial comprised 34.5% of the EO. Generally, some discrepancies among EO compound compositions in publications are apparent due to multiple factors, including geographic localization and temperature [42,43].

It was not surprising that citral showed a characteristic inhibition zone (white spots on Figure 5 and Figure 6) on the TLC-based bioautographic silica plate as this feature has been previously published in the cases of other bacterium species [44]. Our results further support that lemongrass EO is effective against hitherto uncharacterized bacterium species, including the potential aetiological agents of PK. Beyond citral, α-terpineol also induced antibacterial effects. In order to reveal other potential compounds with antibacterial features in lemongrass EO, HS-SPME/GC-MS analysis was used. The SPME method is an anhydrous sampling technique that involves the use of a fibre-coated extraction phase. In contrast to steam distillation, this technique works at a lower temperature, meaning that the compounds do not decompose or transform, and the proportion of more volatile components increases compared to the less volatile components [45]. Unfortunately, in the frame of this study we could not identify if γ-cadinene and calamenene were antibacterial as these compounds were not available from any supplier. We could only speculate if these EO compounds could also contribute to the antibacterial feature represented by a white spot on the TLC plate (Figure 5C(4),D(4) and Figure 6C(4)) or not. Further studies addressing this issue in the future could clarify this question.

The low MIC and MBC values (0.1 mg/mL) of all three tested bacteria (Table 4) confirmed the strong antibacterial activity of citral, similar to recent findings [35] using *B. cereus* (0.15 mg/mL). Although the antibacterial feature of α-terpineol was less pronounced (Table 3 and Table 4), results from the drop plate method, MIC and MBC tests indicate that this compound—together with cadinene and calamenene—might also contribute to the antibacterial feature of the spot that was analysed by HS-SPME/GC-MS.

Our results highlight the potential of therapeutics based on West Indian lemongrass EO in the treatment of PK and possibly other superficial skin infections, and revealed the presence of specific compounds with potential therapeutic value.

## 4. Materials and Methods

### 4.1. Bacterium Isolates and Growth Conditions

*Bacillus thuringiensis* was isolated from the PK lesions of a 43-year-old man that presented with a prolonged history of malodour and burning irritation of his soles [8]. *Kytococcus sedentarius* (DSM 20547) and *Dermatophilus congolensis* (DSM 44180) were purchased from the German Strain collection (Leibniz Institute, Braunschweig, Germany). All bacteria were routinely grown under aerobic conditions. Tryptic soy agar (TSA) (Oxoid, New York, NY, USA) was used to grow *B. thuringiensis* (37 °C) while blood agar (BA) was used for *D. congolensis* (37 °C) and *K. sedentarius* (30 °C). Tryptic soy broth (TSB) (Oxoid, New York, NY, USA) was used for the minimal inhibitory concentration (MIC) and minimal bactericidal concentration (MBC) determinations.

### 4.2. Essential Oils

Twelve EOs (rosemary (*Rosmarinus officinalis*), lemongrass (*Cymbopogon citratus*), clove (*Eugenia caryophyllata*), salvia (*Salvia sclarea*), cinnamon (*Cinnamomum zeylanicum*), citronella (*Cymbopogon nardus*), eucalyptus (*Eucalyptus globulus*), fennel (*Foeniculum vulgare*), spearmint (*Mentha spicata*), peppermint (*Mentha piperita*), lemon (*Citrus limon*), and thyme (*Thymus vulgaris*)) were tested for their antibacterial effects on the bacterium species. The EOs were directly purchased from the Aromax Co. (Budapest, Hungary).

### 4.3. Antibacterial Testing—Drop Plate Method

The antibacterial effects of the EOs were examined on agar plates by the drop plate method. Prior to the experiments, bacterial cell counts were standardized by setting their optical densities (OD) at 600 nm (OD_600_) to 0.2 (~10^8^ CFU/mL) in PBS. The suspension of *B. thuringiensis* (100 µL) was spread on TSA plates, and 100 µL from *K. sedentarius* and *D. congolensis* was spread on BA plates. After drying, 5 µL of each EO was dropped onto the plate surfaces. The TSA plates were incubated at 37 °C for 24 h and the BA plates with *K. sedentarius* were incubated at 30 °C for 24 h. The following day, the diameters of inhibitions were measured in mm.

The same procedure was applied for the drop plate method of citral and α-terpineol. For controls, the antibiotics erythromycin and clindamycin were used.

### 4.4. Minimum Inhibitory Concentration (MIC) and Minimum Bactericidal Concentration (MBC)

The MIC is defined as the minimum concentration of a drug that prevents visible growth of bacterium, while MBC is defined as the minimum drug concentration that reduces the growth (population) of microbial colonies by at least 99.9% [46]. The MICs and MBCs were determined by macrodilution tests. Bacterium isolates were grown and adjusted to OD_600_ = 0.2 [46]. Serial dilutions were carried out in 5-5 mL of TSB medium. To these volumes, 0.5, 1, 2, 4, 8, 16 and 32 µL of EO extracts were added to obtain concentrations of 0.1, 0.2, 0.4, 0.8, 1.6, 3.2 and 6.4 mg/mL, respectively. The bacterium suspensions (5 µL) were added to the reagent tubes. Control tubes either did not contain bacteria (EO control) or did not contain bacteria and EOs (medium control). All tubes were incubated at 37 °C except *K. sedentarius* (30 °C). Inhibitory and bactericidal effects were determined the next day by running off 10 μL suspensions on TSA plates for *B. thuringiensis,* and after 3 days of incubation on BA plates for *D. congolensis* and *K. sedentarius* from visually clear reagent tubes.

The same procedure was applied for the MIC and MBC determinations of citral and α-terpineol by diluting the pure compounds from 12.8 mg/mL to 0.1 mg/mL. For controls, the antibiotics erythromycin and clindamycin were used.

### 4.5. Spore-Formation Inhibition Test

The ability of EOs to impede sporulation was examined using overnight cultures of *Bacillus thuringiensis*. This culture was diluted 1000 times and left to grow for 3 h (37 °C) to facilitate spore germination. From this culture the CFU was determined, and 5 µL suspensions were added to the test tubes containing 5 mL of TSB medium, followed by different quantities (0.5, 1, 2, 3, 4, 8, 16, 32, 64, 128 and 256 μL) of lemongrass essential oil. The final solutions had concentrations of 0.1, 0.2, 0.4, 0.6, 0.8, 1.6, 3.2, 6.4, 12.5, 25 and 50 mg/mL, respectively. After 24 h of incubation at 37 °C, 1-1 mL from the optically clear tubes was transferred to flasks containing 25 mL of sterile TSB medium. After 48 h of incubation at 37 °C, 10 μL was plated on TSA plates to detect living bacteria. The absence of colonies was considered a clear indicaton that no sporulation has occured and that survival of *B. thuringiensis* was hindered. Experiments were repeated 3 times.

### 4.6. Headspace Solid-Phase Microextraction (SPME) Conditions

The lemongrass EO sample and the white spots from the developed TLC-based bioautography plates were analysed with HS-SPME. In the case of lemongrass EO the liquid EO was analysed, while in the case of the TLC spot the cut-out silica plate was the analysed sample. The samples were put into vials (20 mL) and sealed with a silicon/PTFE septum prior to HS-SPME/GC-MS analysis. Sample preparation using the static headspace solid-phase microextraction (sHS-SPME) technique was carried out with a CTC Combi PAL (CTC Analytics AG, Zwingen, Switzerland) automatic multipurpose sampler using a 65 μM StableFlex carboxen/polydimethylsiloxane/divinylbenzene (CAR/PDMS/DVB) SPME fibre (Supelco, Bellefonte, PA, USA). After an incubation period of 5 min at 100 °C, extraction was performed by exposing the fibre to the headspace of a 20 mL vial containing the sample for 10 min at 100 °C. The fibre was then immediately transferred to the injector port of the GC/MS and desorbed for 1 min at 250 °C. Injections were made in splitless mode. The SPME fibre was cleaned and conditioned in a fibre bakeout station in a pure nitrogen atmosphere at 250 °C for 15 min.

### 4.7. Gas Chromatography–Mass Spectrometry (GC-MS) Analysis

The analyses were carried out with an Agilent 6890N/5973N GC-MSD (Santa Clara, CA, USA) system equipped with a Supelco (Sigma-Aldrich, Philadelphia, PA, USA) SLB-5MS capillary column (30 m × 250 µm × 0.25 µm). The GC oven temperature was programmed to increase from 60 °C (3 min isothermal) to 250 °C at 8 °C/min (1 min isothermal). High purity helium (6.0) was used as a carrier gas at 1.0 mL/min (37 cm/s) in constant flow mode. The mass selective detector (MSD) was equipped with a quadrupole mass analyser and was operated in electron ionization mode at 70 eV in full scan mode (41–500 amu at 3.2 scan/s). The data were evaluated using MSD ChemStation D.02.00.275 software (Agilent, Santa Clara, CA, USA). The identification of the compounds was carried out by comparing retention data and recorded spectra with the literatury data, and the NIST 2.0 library was also consulted. The percentage evaluation was carried out by area normalization.

### 4.8. Thin-Layer Chromatography Combined with Direct Bioautography (TLC-DB)

The overall composition and specific antibacterial compounds of lemongrass EO were visualized in parallel on two preconditioned (100 °C for 30 min) 5 × 10 cm 60 F_254_ TLC plates (Merck, Darmstadt, Germany) as described previously [47], with slight modifications. Aliquots (0.2 μL) were deposited in a horizontal thin line at the bottom of the the plates, and ethanol served as a solvent control. Citral (20 mg/mL; Sigma Technology Hungary, Budapest, Hungary) and α-terpineol (100 mg/mL; Sigma Technology Hungary) were used with known running features. The TLC plates were developed with toluene–ethyl acetate (97:3) in a saturated twin trough chamber (CAMAG, Muttenz, Switzerland). The plate was incubated (at room temperature, 1 h) in 50 mL of TSB in the case of *D. congolensis* and for 10 s in the cases of the *B. thuringiensis* and *K. sedentarius* suspensions (3 × 10^8^ cfu mL^−1^). Then, the plates were incubated in a vapour chamber at 37 °C for 2 h (*B. thuringiensis)* or 6 h (*D. congolensis* at 37 °C and *K. sedentarius* at 30 °C). The following day the plates were immersed in an aqueous solution of 3-(4,5-dimethylthiazol-2-yl)-2,5-diphenyltetrazolium bromide (MTT, 0.05 g/90 mL) for 10 s and incubated under aerobic conditions in a vapour chamber (*B. thuringiensis* and *D. congolensis* at 37 °C, *K. sedentarius* at 30 °C) until the white spots appeared. The antibacterial activities of the separated compounds were indicated by the presence of these white spots against the bluish background [47]. All measurements were performed in duplicate. Separated compounds of lemongrass EO were visualized by dipping one TLC plate into the ethanolic vanillin–sulphuric acid reagent and heated for 5 min at 90 °C. The separated components were characterized according to Rf values, determined by the known standards (citral, α-terpineol) and Kovats index. The definition of the Rf value is the distance travelled by a given component divided by the distance travelled by the solvent front. For a given system at a known temperature, it is a characteristic of the component and helps to identify components [48]. The Kovats retention index is a concept introduced in GC to convert retention time into a more reliable and reproducible system [49].

## 5. Conclusions

The antimicrobial effects of 12 essential oils were confirmed on a group of bacteria responsible for the superficial skin infection pitted keratolysis. Lemongrass proved to be the most effective EO as it was most able to actively inhibit spore formation. From among the compounds, the antibacterial effect of citral was confirmed on all the known aetiological agents, and α-terpineol was revealed to possess a similar effect. Results of this study have demonstrated the potential of therapeutics containing lemongrass EO in the treatment of PK.

## Figures and Tables

**Figure 1 molecules-27-01423-f001:**
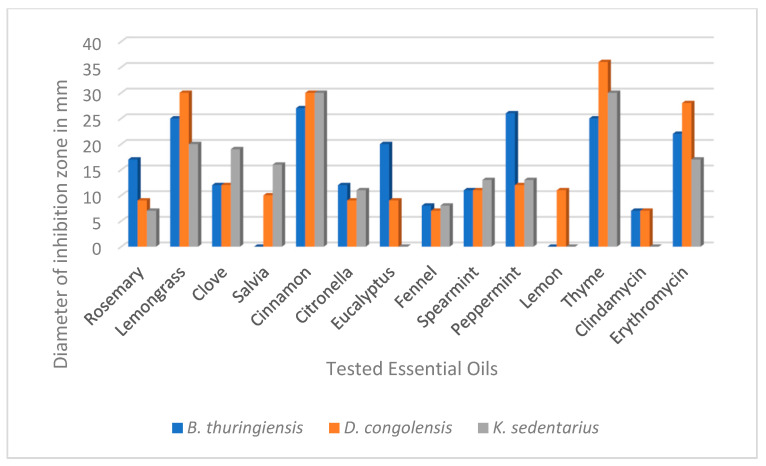
Drop plate antibacterial test results of the 12 essential oils on the three bacterium species associated with pitted keratolysis. Values represent the inhibition zones in diameters, determined around the drops. Clindamycin and erythromycin were used as positive standards (applied concentrations were 2 mg/mL).

**Figure 2 molecules-27-01423-f002:**
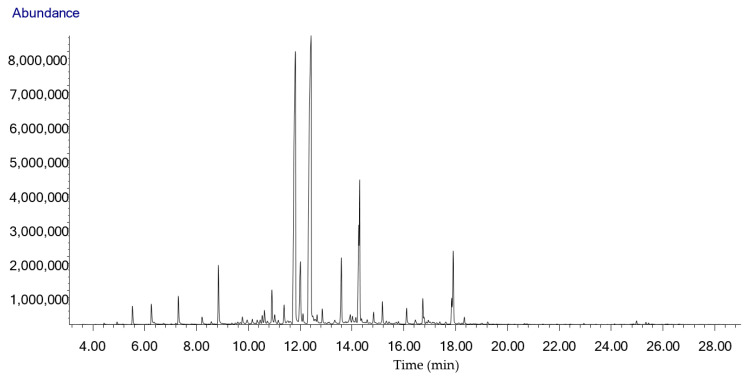
Chromatogram of the HS-SPME/GC-MS analysis of the West Indian lemongrass essential oil (*Cymbopogon citratus*) used in this study obtained from Aromax Ltd. Each compound can be identified based on the data in Table 2 (tR (min)).

**Figure 3 molecules-27-01423-f003:**
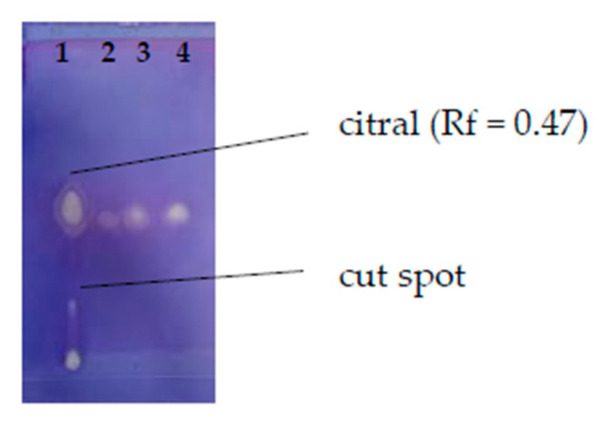
Compounds with antibacterial activities of lemongrass essential oil revealed on *B. thuringiensis* by TLC-DB. Order and quantity of the volatile test materials on the plate was the following: 1-lemongrass EO (0.2 mg); 2-citral (0.02 mg); 3-citral (0.04 mg); 4-citral (0.08 mg).

**Figure 4 molecules-27-01423-f004:**
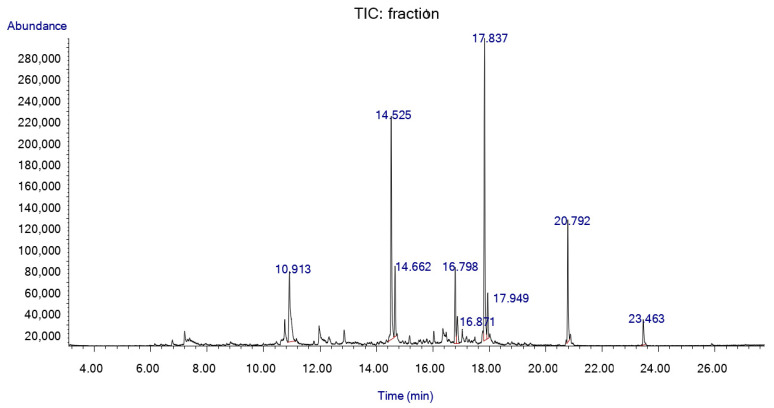
HS-SPME/GC-MS chromatogram of the white cut-out spot from TLC-BD silica plate.

**Figure 5 molecules-27-01423-f005:**
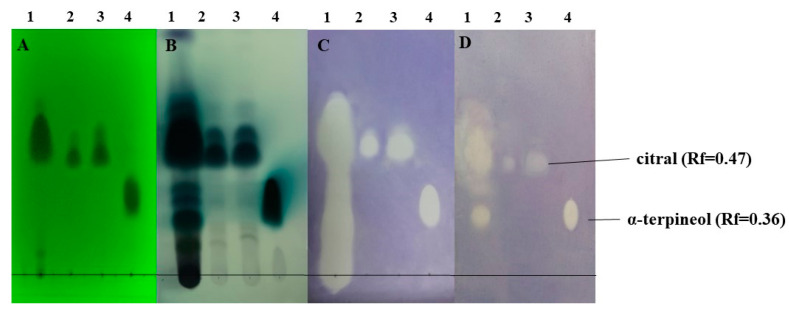
Antibacterial components in the lemongrass EO used in this study after TLC-DB. (**A**) Plate under UV 254 nm, (**B**) TLC plate after treatment with vanillin–sulfuric acid reagent and documented in visible light, (**C**) TLC-DB assay: bioautograms using *B. thuringiensis*, (**D**) TLC-DB assay: bioautograms using *K. sedentarius* (bright zones indicate antibacterial effects). Mobile phases: dichloromethane and toluene–ethyl acetate 93:7 (*v/v*). The applied volumes of compounds used were as follows: lemongrass EO—6 µL, citral—4.5 and 6 µL, α-terpineol—1.5 µL. The stock solutions had concentrations as follows: lemongrass EO—200 mg/mL, citral—20 mg/mL, α-terpineol—100 mg/mL. Order and quantity of the volatile test materials on each plate were as follows: 1-lemongrass EO (1.2 mg); 2-citral (0.09 mg); 3-citral (0.12 mg); 4-α-terpineol (0.15 mg).

**Figure 6 molecules-27-01423-f006:**
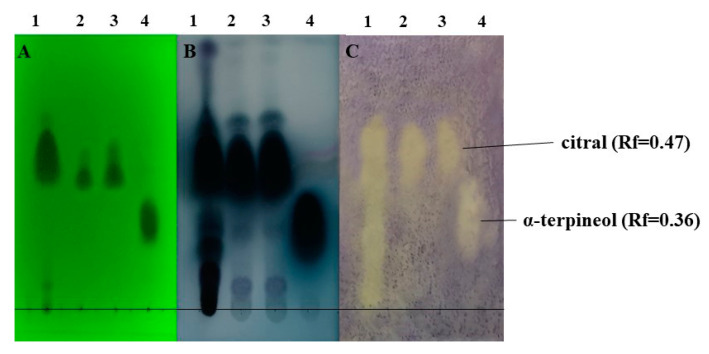
Antibacterial components in the lemongrass EO used in this study after TLC-DB. (**A**) plate under UV 254 nm, (**B**) TLC plate after treatment with vanillin–sulfuric acid reagent and documented in visible light, (**C**) TLC-DB assay: bioautograms using *D. congolensis* (bright zones indicate antibacterial effects). Mobile phases: dichloromethane and toluene–ethyl acetate 93:7 (*v/v*); 1, 4.5 and 6 µL indicated the applied volumes of the EO and the standards (lemongrass essential oil: 1µL, citral: 4.5 and 6 µL, α-terpineol: 4.5 µL). The lemongrass essential oil was applied without diluton and the stock solution in the cases of citral and α-terpineol were 100 mg/mL. Order and quantity of the volatile test materials on each plate were as follows: 1-lemongrass essential oil (undiluted); 2-citral (0.45 mg); 3-citral (0.6 mg); 4-α-terpineol (0.45 mg).

**Table 1 molecules-27-01423-t001:** Minimal inhibitory and bactericidal concentrations (MIC and MBC) of the 12 essential oils on the three bacterium species associated with pitted keratolysis. MIC and MBC values are presented in mg/mL. Clindamycin and erythromycin were used as positive standards. Values are presented in mg/mL.

	*B. thuringiensis*		*D. congolensis*		*K. sedentarius*	
	MIC	MBC	MIC	MBC	MIC	MBC
Rosemary	0.5 ± 0.02	0.5 ± 0.02	0.6 ± 0.07	0.6 ± 0.02	0.2 ± 0	0.5 ± 0.03
Lemongrass	0.15 ± 0	0.2 ± 0	0.15 ± 0	0.15 ± 0.02	0.1 ± 0	0.15 ± 0
Clove	0.3 ± 0.04	0.4 ± 0	0.2 ± 0	0.2 ± 0	0.1 ± 0	0.25 ± 0.04
Salvia	0.3 ± 0.04	0.5 ± 0.04	0.5 ± 0.03	0.5 ± 0	0.1 ± 0	0.1 ± 0
Cinnamon	0.1 ± 0	0.15 ± 0	0.1 ± 0	0.1 ± 0	0.1 ± 0	0.1 ± 0
Citronella	0.3 ± 0.04	0.4 ± 0	0.3 ± 0	0.3 ± 0	0.15 ± 0	0.9 ± 0.07
Eucalyptus	1.2 ± 0.16	1.2 ± 0.26	2.4 ± 0.13	2.4 ± 0.03	1.8 ± 0.07	2.4 ± 0.03
Fennel	4 ± 0.39	4.8 ± 0.26	0.8 ± 0	0.8 ± 0	0.3 ± 0.04	3.3 ± 0.05
Spearmint	0.6 ± 0.08	0.6 ± 0.08	0.6 ± 0.08	0.6 ± 0.08	0.4 ± 0	0.8 ± 0
Peppermint	0.4 ± 0	0.4 ± 0	0.6 ± 0	0.6 ± 0.06	0.25 ± 0.01	0.6 ± 0
Lemon	1.2 ± 0.16	1.2 ± 0.16	1.2 ± 0.02	2 ± 0	0.45 ±0.03	1 ± 0.02
Thyme	0.1 ± 0	0.1 ± 0	0.1 ± 0	0.1 ± 0	0.1 ± 0	0.1 ± 0
Clindamycin	0.8 ± 0	0.8 ± 0	0.1 ± 0	0.1 ± 0	0.2 ± 0	0.8 ± 0
Erythromycin	0.1 ± 0	0.1 ± 0	0.1 ± 0	0.1 ± 0	0.1 ± 0	0.1 ± 0

**Table 2 molecules-27-01423-t002:** Percentage compound composition of the West Indian lemongrass essential oil (*Cymbopogon citratus*) used in this study. Compounds determined from the outcutted from the TLC-BD plate are labelled with bold.

Name of Compounds	tR (min)	KI	Presence in %	
			Lemongrass EO	Fraction
Camphene	5.5	941	0.8	
Limonene	7.3	1023	1.3	
Linalool	8.8	1091	2.6	
Verbenol	10.5	1171	0.4	
**α-Terpineol**	**10.9**	**1190**	**1.7**	**13.2**
Carveol, trans	11.0	1195	0.5	
Carveol, cis	11.4	1216	1.0	
Neral	11.8	1237	26.1	
Geraniol	12.0	1247	4.3	
Piperitone	12.1	1253	0.6	
Geranial	12.4	1268	34.5	
Geranyl formate	12.8	1289	0.7	
Neryl acetate	13.9	1350	0.7	
β-Caryophyllene	15.2	1424	1.0	
**Cadinene**	**16.7**	**1513**	**1.4**	**7.0**
**Calamenene**	**16.9**	**1525**	**0.6**	**3.7**
Caryophyllene-oxide	17.9	1588	4.7	
Sum			82.2	
M^+^140	6.3	977	0.8	
M^+^155	8.2	1064	0.4	
M^+^152	9.7	1133	0.5	
M+166	10.6	1176	0.6	
M^+^166	12.6	1279	0.4	
M^+^168	13.6	1333	3.2	
M^+^164	14.1	1361	0.4	
M^+^204	14.3	1372	9.8	
M^+^166	14.8	1400	0.6	
M^+^204	16.1	1475	0.7	
Sum			17.4	

**Table 3 molecules-27-01423-t003:** Antibacterial effects of citral and α-terpineol (5 μL) on the three tested bacterium species. Values represent the diameters of the inhibition zones in mm.

	Citral (Diameter in mm)	α-Terpineol (Diameter in mm)
** *B. thuringiensis* **	29 ± 2	17 ± 3
** *D. congolensis* **	23 ± 2	7 ± 1
** *K. sedentarius* **	14 ± 2	6 ± 1

**Table 4 molecules-27-01423-t004:** Minimal inhibitory and minimal bactericidal concentrations of citral and α-terpineol tested with the macrodilution method. Concentrations are presented in mg/mL.

	Citral		α-Terpineol	
	MIC	MBC	MIC	MBC
** *B. thuringiensis* **	0.1 ± 0	0.1 ± 0	0.8 ± 0	0.8 ± 0
** *D. congolensis* **	0.1 ± 0	0.1 ± 0	0.4 ± 0	0.8 ± 0
** *K. sedentarius* **	0.1 ± 0	0.1 ± 0	0.4 ± 0	0.8 ± 0

## Data Availability

Not applicable.

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
