# Peer review of "Antibacterial Effect of Lemongrass (Cymbopogon citratus) against the Aetiological Agents of Pitted Keratolyis"

_molecules, 2022, doi:10.3390/molecules27041423_

Round 1

Reviewer 1 Report

The antibacterial effects of 12 EOs were compared in the first part of this study against the three known aetiological agents of PK (Kytococcus sedentarius, Dermatophilus congolensis and Bacillus thuringiensis). Lemongrass EO was further chose for testing against all three bacterium species and further analyzed. Thin layer chromatography combined with direct bioautography (TLC-BD) was used to detect the presence of antibacterial compounds from lemongrass EO. The TLC spot that was not citral was separated and further analyzed by HS-SPME/GC-MS. The manuscript contains interesting parts regarding the observed biological activities, but the part regarding the chemical analysis of lemongrass essential oil is very poor and it needs major improvement. In fact, the present isomers in the analyzed oil should be better determined using the comparison of the obtained retention indices. In addition, there is need to point out very clearly what is the novelty of present research? This is very crucial part, because lemongrass essential oil is very well known and there is need to provide detail explanation about the novelty of the research. English grammar should be rechecked throughout overall manuscript and improved. Therefore the manuscript needs major reconsideration.

Particular remarks:

Abstract “Citral showed a characteristic spot at the Rf value of 0.47, while the solid phase microextraction (SPME) analysis of an unknown spot with strong antibacterial activity has revealed the presence of α- terpineol, cadinene and calamenene.” – There is need to explained more clearly how the spot was separated and analyzed by HS-SPME/GC-MS. The headspace “HS” is crucial to be pointed out in the abstract as well as through the introduction, goals, results and discussion part of the manuscript.

Lines 57-59: “They are typically characterized by two or three major components at relatively high concentrations (20– 58 70%) compared to others components present in trace amounts [11].”. Although the authors cite the reference 11 it is not true that EO are typically characterized by 2 or 3 compounds with high concentrations. It all depends on the type of the essential oil. For example, the essential oil from Helichrysum species is more complex. Therefore, the authors should avoid exact number of major compounds in the oils and it is better to mention “several compounds”.

Lines 75-78: “Spots with potential antibacterial activities were cut out from the TLC-DB plates and 75 separated by solid phase microextraction (SPME). We then identified citral and alpha- 76 terpineol as active components against the all the known aetiological agents of PK.” – These are the results of the study, that is not appropriate for the introduction part and those sentences should be removed or modified. The authors should state here the novelty of present research (what was done for the first time?) and to present the research hypothesis as well how is it to be tested in the research?

The Rt values are redundant from Table 2 and they should be removed. In addition, the nomenclature of carveol should be corrected to cis-carveol and trans-carveol. What cadinene isomer is present?

The legend of figure 3 should be corrected: 1-lemongrass essential oil instead of 1-lemongrass. In the same way, there is need to correct the legends of all other figures. Also within the overall manuscript manuscript there is need always to write lemongrass essential oil, not only lemongrass.

Lines 153-154: “…while the rest represented solvents used for separation.” – This part should be removed from the manuscript. Consequently, Table 3 should be corrected to contain only identified essential oil components, not the used solvents. In addition, retention indices should be reported in the table, not the retention times.

In the discussion part there is need to elaborate detail why from all tested essential oils (rosemary, lemongrass, clove, salvia, cinnamon, citronella, eucalyptus, fennel, spearmint, peppermint, lemon and thyme) the author choose lemongrass essential oil for further study? It is not adequate to justify the selection of lemograss essential oil with “From among the most effective bactericidal EOs (thyme, cinnamon and lemongrass), this latter one was chosen for further studies.” Why lemongrass essential oil? Broad testing of different essential oils and selection of one for further study should be clearly emphasized at the end of introduction part within the research design of hypothesis.

Lines 283-285: “This hypothesis is supported by a recent observations where the antibacterial efficacies of 7-Hydroxycalamenene [46] and δ-cadinene [47] were demonstrated.” – The statement is not appropriate for current discussion because the structure of 7-hydroxycalamenene is significantly different than calamenene and OH group could significantly influence the bioactivity and it should be removed. In addition, there are very well known differences in the isomers biological activities and δ-cadinene was not determined in present research. In fact, it would be very useful if the authors determine what isomer is present based on the determine retention index.

In the experimental part there is need to explain how GC-MS analysis was performed for lemongrass essential oil (the results presented in Table 2)! Was this direct analysis by the injection of dissolvent essential oil in the solvent, or it was HS-SPME/GC-MS analysis?

The conclusion should be completely changed and significantly improved. Namely, the first part of conclusion is containing know data not directly connected with the present research that should be omitted or shortened. The second part is containing general conclusion of performed research, but there is need to add much more concrete details and findings from the obtained results referring to concrete compounds and the results.

Author Response

Detailed Responses to Editor and Reviewers

Manuscript ID: Molecules-1575942

Response to the Editor’s comments

We are very pleased to resubmit for publication the revised version of “Antibacterial effect of lemongrass (Cymbopogon citratus) against the aetiological agents of pitted keratolyis” (Microorgagisms-1575942) Bettina Schweitzer, Viktória Lilla Balázs, Szilárd Molnár, Bernadett Szögi-Tatár, Andrea Böszörményi, Tamás Palkovics, Györgyi Horváth, György Schneider to be considered for publication as an original article in MDPI Molecules. We carefully considered your comments hoping our revision has improved the paper to a level of your satisfaction.

Again, we appreciate the opportunity to revise our work for consideration for publication in MDPI Molecules.

The authors thank the reviewers for their rapid and constructive reviews of the manuscript. Here are our detailed responses to the reviewer's issues. Reviewers comments are reported in red.

Response to Reviewer #1 Comments

The antibacterial effects of 12 EOs were compared in the first part of this study against the three known aetiological agents of PK (Kytococcus sedentarius, Dermatophilus congolensis and Bacillus thuringiensis). Lemongrass EO was further chose for testing against all three bacterium species and further analyzed. Thin layer chromatography combined with direct bioautography (TLC-BD) was used to detect the presence of antibacterial compounds from lemongrass EO. The TLC spot that was not citral was separated and further analyzed by HS-SPME/GC-MS. The manuscript contains interesting parts regarding the observed biological activities, but the part regarding the chemical analysis of lemongrass essential oil is very poor and it needs major improvement. In fact, the present isomers in the analyzed oil should be better determined using the comparison of the obtained retention indices. In addition, there is need to point out very clearly what is the novelty of present research? This is very crucial part, because lemongrass essential oil is very well known and there is need to provide detail explanation about the novelty of the research. English grammar should be rechecked throughout overall manuscript and improved. Therefore the manuscript needs major reconsideration.

Response: First of all thank You for your critical review. We tried to do our best to improve the quality of the work accordingly.

Remark:

Abstract “Citral showed a characteristic spot at the Rf value of 0.47, while the solid phase microextraction (SPME) analysis of an unknown spot with strong antibacterial activity has revealed the presence of α- terpineol, cadinene and calamenene.” – There is need to explained more clearly how the spot was separated and analyzed by HS-SPME/GC-MS. The headspace “HS” is crucial to be pointed out in the abstract as well as through the introduction, goals, results and discussion part of the manuscript.

Response: Thank you for your remark. We corrected SPME to HS-SPME/GC-MS through out the text.

Remark:

Lines 57-59: “They are typically characterized by two or three major components at relatively high concentrations (20– 58 70%) compared to others components present in trace amounts [11].”. Although the authors cite the reference 11 it is not true that EO are typically characterized by 2 or 3 compounds with high concentrations. It all depends on the type of the essential oil. For example, the essential oil from Helichrysum species is more complex. Therefore, the authors should avoid exact number of major compounds in the oils and it is better to mention “several compounds”.

Response: We have modified the two relevant sentences accordingly as it was true for lemongrass EO, yes but sure might not true for other EOs.  (L.57)

Remark:

Lines 75-78: “Spots with potential antibacterial activities were cut out from the TLC-DB plates and 75 separated by solid phase microextraction (SPME). We then identified citral and alpha- 76 terpineol as active components against the all the known aetiological agents of PK.” – These are the results of the study, that is not appropriate for the introduction part and those sentences should be removed or modified. The authors should state here the novelty of present research (what was done for the first time?) and to present the research hypothesis as well how is it to be tested in the research?

Response: Thank You for reasonable remark. We modified this section and emphasized the novelty of this work with a potential practical relevance. (L. 76-85)

Remark:

The Rt values are redundant from Table 2 and they should be removed. In addition, the nomenclature of carveol should be corrected to cis-carveol and trans-carveol. What cadinene isomer is present?

Response: We removed the Rt values from Table 2. and we clarified the nomenclature. γ-cadinene was present in the West Indian lemongrass essential oil.

Remark:

The legend of figure 3 should be corrected: 1-lemongrass essential oil instead of 1-lemongrass. In the same way, there is need to correct the legends of all other figures. Also within the overall manuscript manuscript there is need always to write lemongrass essential oil, not only lemongrass.

Response: Thank you for your reasonable commence. We corrected them.

Remark:

Lines 153-154: “…while the rest represented solvents used for separation.” – This part should be removed from the manuscript. Consequently, Table 3 should be corrected to contain only identified essential oil components, not the used solvents. In addition, retention indices should be reported in the table, not the retention times.

Response: The relevant part of the sentence was removed. Solvents came from the running buffer that was applied for separation of the samples on the TLC plates (Line 166). We took Table 3 out from the manuscript.  

Remark:

In the discussion part there is need to elaborate detail why from all tested essential oils (rosemary, lemongrass, clove, salvia, cinnamon, citronella, eucalyptus, fennel, spearmint, peppermint, lemon and thyme) the author choose lemongrass essential oil for further study? It is not adequate to justify the selection of lemograss essential oil with “From among the most effective bactericidal EOs (thyme, cinnamon and lemongrass), this latter one was chosen for further studies.” Why lemongrass essential oil? Broad testing of different essential oils and selection of one for further study should be clearly emphasized at the end of introduction part within the research design of hypothesis.

Response: This point we referred in the discussion part (Lines 254-260).

Remark:

“This hypothesis is supported by a recent observations where the antibacterial efficacies of 7-Hydroxycalamenene [46] and δ-cadinene [47] were demonstrated.” – The statement is not appropriate for current discussion because the structure of 7-hydroxycalamenene is significantly different than calamenene and OH group could significantly influence the bioactivity and it should be removed. In addition, there are very well known differences in the isomers biological activities and δ-cadinene was not determined in present research. In fact, it would be very useful if the authors determine what isomer is present based on the determine retention index.

Response: Your remark concerning to the correlation between the position of a group and the bioactivity of the whole molecule is evident. We only wanted to demonstrate that we only could find data about the isomers, but in order to avoid misundertsandings we deleted the relevant sentence and also the references. (Lines 283-286) 

Remark:

In the experimental part there is need to explain how GC-MS analysis was performed for lemongrass essential oil (the results presented in Table 2)! Was this direct analysis by the injection of dissolvent essential oil in the solvent, or it was HS-SPME/GC-MS analysis?

Response: In all cases our method was the HS-SPME/GC-MS. In this case no sample pretreatment and no solvent was necessary. The sample was put in a small vial that was warmed up and from the headspace the SPME fibre took up the volatile compounds. This fibre got into the GC and compounds were detached from the fibre by heating. After that the sample was injected and analysed on the column of GC. This we have inserted into 4.6 (L. 358-361).

Remark:

The conclusion should be completely changed and significantly improved. Namely, the first part of conclusion is containing know data not directly connected with the present research that should be omitted or shortened. The second part is containing general conclusion of performed research, but there is need to add much more concrete details and findings from the obtained results referring to concrete compounds and the results.

Response: You critic is reasonable. We reformulated the Conclusions and tried to grab now novelties also from the microbiological points and correlating analytics.  (L. 411-417)

Thank You again for your help and commences!

We hope that these revisions improve the paper such that you and the reviewers now deem it worth of publication in MDPI Molecules.

Reviewer 2 Report

Manuscript Molecules 157594

The manuscript submitted by Schweitzer et al reports the screening of twelve essential oils isolated from aromatic plants such as lemongrass, clove, and eucalyptus for their inhibitory potential over three bacterium species that are among the etiological agents associated with pitted keratolysis. This skin condition can become chronic and generates concern mainly in hospitalized people or residents of places subject to constant flooding. Additionally, some professionals are subject to pitted keratolysis, since this condition is associated with the frequent use of footwear, which occurs with athletes and soldiers for example. In this way, the problem is of comprehensive scientific and social interest. The essential oils of thyme, cinnamon and lemongrass were the most active against the test microorganisms.

In sequence, the authors continued the work with lemongrass essential oil, evaluating the performance of this oil in the formation of spores by Bacillus thuringiensis, the composition and other features of this essential oil.

The purpose of the work is confusing. Although the authors made a screening, usually used to define the most active sample, the focused essential oil was not the most active one. In Line 71, the authors explain that they followed the work with lemongrass due to results they had (but did not present) and the "recent" results presented in the literature (reference 18, which is from 2009, so it is definitely not recent; this should be corrected). In line 71 and in the discussion, the reason for choosing lemon grass oil should be clearly justified, not only in relation to previous results but also in relation to the screening result. Something like that: Among the most active EOs, a detailed study was conducted with lemongrass. Although it was not the most active one…..

I am very uncomfortable with the contents of lines 78-81. A statement like this in an article in a respected scientific journal as Molecules could be used to promote the trade of essential oil-based products for the treatment of skin infections without proper medical control. The use of antibiotics causes numerous problems and is a challenge, but the research presented in the manuscript is not deepen enough to suggest that essential oils may dispense the use of antibiotics. Another highly questionable point is the fact that the authors consider that essential oils have no antibiotic effect, since they suggest that they can replace antibiotics. I cannot understand what activity the EO have if not antibiotic. Still, the results presented provide no indication that essential oils avoid the phenomenon of resistance. Therefore, in addition to the following observations, in my opinion, the manuscript cannot be approved for publication without a profound modification in this passage and in other places where assumptions like this are made.

Results Section (refering also to Methods)

The results presented in Figure 1 are impressive, because drop plate method has the great drawback of not fitting well to lipophilic samples, which is the case for essential oils. These compounds do not spread well in the agar (aqueous medium, i.e. polar) and the inhibition zones are not proportional to the inhibition capacity in general. In addition, essential oils are volatile and their concentration in the spot decreases during incubation. To validate these good results, it is necessary to use a positive standard so that the value of inhibition halos is compared.

In case the EO were water soluble, the authors should specify this information, which would better justify the use of drop plate method. Similarly, for the MIC and MBC assay, a bi-phasic system would be expected when mixing an essential oil with the aqueous culture medium. Did the authors use any emulsifying agents? Or, again, were the oils water-soluble? Here also the positive control is missing.

GCMS Analysis

The determination of lemnograss EO components is well studied, so the data presented are not new. Still, 10 components were not identified (line 121). Why is it? The unidentified components add up to 17.1% according to Table 2 and may be correlated with the activity of the EO. In addition, the authors describe that they did a GCMS analysis but explain that EOs identification was made by comparison with standards and calculations using Kovats retention index. Thus, it is not clear whether mass spectra were actually obtained for identification. If they have not used it, the analysis did not occur by GCMS and the description of the methodology (and other parts) that mention the use of MS should be corrected. If they have used MS, they should be able to identify all components. If they don’t, the mass spectra of the unidentified components must be attached as supplementary material.

Line 122. Figure 1. Please, provide a readable spectrum.

Line 153. Why did you use six solvents for extraction? Please, add a reference because it is very unusual.

Line 181. The TLC-DB assay is qualitative and unnecessary when using the MIC assay. However, the results, as the authors point out in lines 186-187, showed a discrete spot with rf lower than citral’s inhibited bacterial growth. That is why I mentioned earlier the importance of identifying the minor components of the extract. As the authors did previously, this spot could be retrieved and analyzed by GCMS for identification, bringing, perhaps, novelty to the work.

Discussion

This section is very long.

I suggest that the authors review when they refer to recent articles, as in line 225, where articles from 10 years ago are cited as recent.

In line 254, a 1982 article is cited to exemplify prescription drugs "today". The authors need to update this information and add information from the really recent literature on the new drugs currently in use or under test for the treatment of PK.

Line 269. When I look at the plates of figure 5, terpineol has a much more defined inhibition spot than citral and, in Figure 6, the plate has no proper definition but, still, it cannot be stated that citral showed the most characteristic inhibition zone. Please, rephrase.

Materials and Methods

Lines 305 and 306. Please, add a space between the number and the unity. Please, check the whole manuscript for this mistake.

Add the names of the controls used in each assay.

Lines 327-329. MIC and MBC are official parameters and have official definitions, which -as I remember- are different from the definition used by the authors. I recommend to refer to the official definition (CLSI for example). If you decide to evaluate your results using LC90, it is ok, but MIC is traditionally LC100. Rephrase, please.

The whole manuscript deserves a deep revision. From line 415-492, authors should remove the instructions from the template

Author Response

Detailed Responses to Editor and Reviewers

Manuscript ID: Molecules-1575942

Response to the Editor’s comments

We are very pleased to resubmit for publication the revised version of “Antibacterial effect of lemongrass (Cymbopogon citratus) against the aetiological agents of pitted keratolyis” (Microorgagisms-1575942) Bettina Schweitzer, Viktória Lilla Balázs, Szilárd Molnár, Bernadett Szögi-Tatár, Andrea Böszörményi, Tamás Palkovics, Györgyi Horváth, György Schneider to be considered for publication as an original article in MDPI Molecules. We carefully considered your comments hoping our revision has improved the paper to a level of your satisfaction.

Again, we appreciate the opportunity to revise our work for consideration for publication in MDPI Molecules.

The authors thank the reviewers for their rapid and constructive reviews of the manuscript. Here are our detailed responses to the reviewer's issues. Reviewers comments are reported in red.

Response to Reviewer #2 Comments

The manuscript submitted by Schweitzer et al reports the screening of twelve essential oils isolated from aromatic plants such as lemongrass, clove, and eucalyptus for their inhibitory potential over three bacterium species that are among the etiological agents associated with pitted keratolysis. This skin condition can become chronic and generates concern mainly in hospitalized people or residents of places subject to constant flooding. Additionally, some professionals are subject to pitted keratolysis, since this condition is associated with the frequent use of footwear, which occurs with athletes and soldiers for example. In this way, the problem is of comprehensive scientific and social interest. The essential oils of thyme, cinnamon and lemongrass were the most active against the test microorganisms.

In sequence, the authors continued the work with lemongrass essential oil, evaluating the performance of this oil in the formation of spores by Bacillus thuringiensis, the composition and other features of this essential oil.

Remark:

The purpose of the work is confusing. Although the authors made a screening, usually used to define the most active sample, the focused essential oil was not the most active one. In Line 71, the authors explain that they followed the work with lemongrass due to results they had (but did not present) and the "recent" results presented in the literature (reference 18, which is from 2009, so it is definitely not recent; this should be corrected). In line 71 and in the discussion, the reason for choosing lemon grass oil should be clearly justified, not only in relation to previous results but also in relation to the screening result. Something like that: Among the most active EOs, a detailed study was conducted with lemongrass. Although it was not the most active one…..

Response: Thank you for your critical view. So we reformulated the relevant parts, both in Introduction and Discussion. (Lines: 70-75 and 253-259)

Remark:

I am very uncomfortable with the contents of lines 78-81. A statement like this in an article in a respected scientific journal as Molecules could be used to promote the trade of essential oil-based products for the treatment of skin infections without proper medical control. The use of antibiotics causes numerous problems and is a challenge, but the research presented in the manuscript is not deepen enough to suggest that essential oils may dispense the use of antibiotics. Another highly questionable point is the fact that the authors consider that essential oils have no antibiotic effect, since they suggest that they can replace antibiotics. I cannot understand what activity the EO have if not antibiotic. Still, the results presented provide no indication that essential oils avoid the phenomenon of resistance. Therefore, in addition to the following observations, in my opinion, the manuscript cannot be approved for publication without a profound modification in this passage and in other places where assumptions like this are made.

Response:

Yes, we our sentences might have been misunderstandable, but it was not our aim to promote essential oils. As spread of antimicrobial resistance is an important issue, we only wanted flash up this option in the future, but certainly fully agree also with that, that proper studies/trials should first clarify their efficacies as potential theraputics in form of ointments or gels. But yes actually the topic of this manuscript does not focus on that.

To reflect your other remark, yes, essential oils can kill bacteria so they are also antibiotics, but our aim with this was to demonstrate that they do not belong to the „traditional antibiotics”. In order to manage your critics, we reformulated this part accordingly and completely reformulated the last paragraph of introduction.

Results Section (refering also to Methods)

The results presented in Figure 1 are impressive, because drop plate method has the great drawback of not fitting well to lipophilic samples, which is the case for essential oils. These compounds do not spread well in the agar (aqueous medium, i.e. polar) and the inhibition zones are not proportional to the inhibition capacity in general. In addition, essential oils are volatile and their concentration in the spot decreases during incubation. To validate these good results, it is necessary to use a positive standard so that the value of inhibition halos is compared.

Remark:

In case the EO were water soluble, the authors should specify this information, which would better justify the use of drop plate method. Similarly, for the MIC and MBC assay, a bi-phasic system would be expected when mixing an essential oil with the aqueous culture medium. Did the authors use any emulsifying agents? Or, again, were the oils water-soluble? Here also the positive control is missing.

Response:

Thank You for your positive but critical remarks. We did not use any emulsifier for MIC and MBC determinations. Certain studies use them (e.g. Tween20 or propyleneglycol), while others not. Forexample Gende et al. (2008) tested the antimicrobial activity of cinnamon oil and the oil was emulsified with propylene glycol. Contrarily, Zulfa et al. (2016) didn’t use any agents in their experiments.

Relating to the the controls, we have repeated the experiments with antibiotic controls, like clindamycin and erithromycin, two classical antibiotics prescribed for PK infections. Results Figure 1 (drop plate method), and Table 1 (MIC, MBC) now contains these controls.  (Figure 1., Table 1., L. 324-325, L. 341-342)

GCMS Analysis

Remark:

The determination of lemnograss EO components is well studied, so the data presented are not new. Still, 10 components were not identified (line 121). Why is it? The unidentified components add up to 17.1% according to Table 2 and may be correlated with the activity of the EO. In addition, the authors describe that they did a GCMS analysis but explain that EOs identification was made by comparison with standards and calculations using Kovats retention index. Thus, it is not clear whether mass spectra were actually obtained for identification. If they have not used it, the analysis did not occur by GCMS and the description of the methodology (and other parts) that mention the use of MS should be corrected. If they have used MS, they should be able to identify all components. If they don’t, the mass spectra of the unidentified components must be attached as supplementary material.

Response:

All the components were identified according to their mass spectrum. One option for the undefineable compounds can be that they are small decomposed components.

Line 122. Figure 1. Please, provide a readable spectrum.

Response: Thank you for your commence. Yes we are also aware that the values are too crwoded and by this not visible, therefore we deleted the values and refered on them in Table 2., according to that the exact values can be identified. (L. 142)

Line 153. Why did you use six solvents for extraction? Please, add a reference because it is very unusual.

Response: These are the solvents that were used for TLC separation not for the Gas chromatography as it was solvent free. In order not to be disturbing for the readers we omit this Table.

Line 181. The TLC-DB assay is qualitative and unnecessary when using the MIC assay. However, the results, as the authors point out in lines 186-187, showed a discrete spot with rf lower than citral’s inhibited bacterial growth. That is why I mentioned earlier the importance of identifying the minor components of the extract. As the authors did previously, this spot could be retrieved and analyzed by GCMS for identification, bringing, perhaps, novelty to the work.

Response: TLC-DB is typically used to identify biologically active compounds, separated on TLC. One spot was cut out from that the presence of 3 compounds were determined and we also want to do this with other compounds, but for that the quality of separation will be increased.      

Discussion

Remark:

This section is very long.

Response:

Thank you for your remark, we cut some microbiological aspects and by this made the manuscript shorter.

Remark:

I suggest that the authors review when they refer to recent articles, as in line 225, where articles from 10 years ago are cited as recent.

Response: We have corrected the term „recent” to „earlier”. (Line 232)

Remark:

In line 254, a 1982 article is cited to exemplify prescription drugs "today". The authors need to update this information and add information from the really recent literature on the new drugs currently in use or under test for the treatment of PK.

Response:

This relevant part is in that microbiological section that we omit from the manuscript and teherefor this question is solved through this.

Remark:

Line 269. When I look at the plates of figure 5, terpineol has a much more defined inhibition spot than citral and, in Figure 6, the plate has no proper definition but, still, it cannot be stated that citral showed the most characteristic inhibition zone. Please, rephrase.

Response:

Yes we have modified it as really can be misleading. (Line 271)

Materials and Methods

Remark:

Lines 305 and 306. Please, add a space between the number and the unity. Please, check the whole manuscript for this mistake.

Response:

Thank you for your commence, we did it and checked through the manuscript.

Remark:

Add the names of the controls used in each assay.

Response:

Controls were added to the text and also to the figures and tables. (Figure 1., Table 1., L. 324-325, L. 341-342)

Remark:

Lines 327-329. MIC and MBC are official parameters and have official definitions, which -as I remember- are different from the definition used by the authors. I recommend to refer to the official definition (CLSI for example). If you decide to evaluate your results using LC90, it is ok, but MIC is traditionally LC100. Rephrase, please.

Response:

Thank You for your clarification. We reformulated the relevant sentence. (L. 327-329)

Remark:

The whole manuscript deserves a deep revision. From line 415-492, authors should remove the instructions from the template

Response:

The relevant part is revised now, and instructions are removed now.

Thank You again for your help and commences!

We hope that these revisions improve the paper such that you and the reviewers now deem it worth of publication in MDPI Molecules.

Round 2

Reviewer 1 Report

The authors revised manuscript according to the comments.

Reviewer 2 Report

It is very good when the authors understand the problems pointed by the referees and modify the manuscript according to the suggestions we give. This was the case of the current manuscript. I am satisfied  with the modifications and I would suggest to accept the manuscript in its current form.